# Functional outcome in home health: Do racial and ethnic minority patients with dementia fare worse?

Jinjiao Wang[1]*, Fang Yu[2], Xueya Cai[3], Thomas V. Caprio[4,5,6], Yue Li[7]

**1** School of Nursing, University of Rochester, Rochester, NY, United States of America, **2** School of Nursing, University of Minnesota, Minneapolis, MN, United States of America, **3** Department of Biostatistics and Computational Biology, University of Rochester, Rochester, NY, United States of America, **4** Department of Medicine, University of Rochester Medical Center, Rochester, NY, United States of America, **5** University of Rochester Medical Home Care, Rochester, NY, United States of America, **6** Finger Lakes Geriatric Education Center, Rochester, NY, United States of America, **7** Department of Public Health Sciences, University of Rochester, Rochester, NY, United States of America

* Jinjiao_wang@urmc.rochester.edu

**Data Availability Statement:** This study utilized existing data from the Outcome and Assessment Information Set (OASIS) and Medicare claims (HH claims) billing records of a non-profit HH agency in

## Abstract

### Objectives

Evaluate the independent and interactive effects of dementia and racial/ethnic minority status on functional outcomes during a home health (HH) admission among Medicare beneficiaries.

### Methods

Secondary analysis of data from the Outcome and Assessment Information Set [OASIS] and billing records in a non-profit HH agency in New York. Participants were adults ≥ 65 years old who received HH in CY 2017 with OASIS records at HH admission and HH discharge. Dementia was identified by diagnosis (ICD-10 codes) and cognitive impairment (OASIS: M1700, M1710, M1740). We used OASIS records to assess race/ethnicity (M0140) and functional status (M1800-M1870 on activities of daily living [ADL]). Functional outcome was measured as change in the composite ADL score from HH admission to HH discharge, where a negative score means improvement and a positive score means decline.

### Results

The sample included 4,783 patients, among whom 93.9% improved in ADLs at HH discharge. In multivariable linear regression that adjusted for HH service use and covariates ($R^2 = 0.23$), being African American (β = 0.21, 95% confidence interval [CI]: 0.06, 0.35, p = 0.005) and having dementia (β = 0.51, 95% CI: 0.41, 0.62, p<0.001) were independently related to less ADL improvement at HH discharge, with significant interaction related to further decrease in ADL improvement. Relative to white patients without dementia, African American patients with dementia (β = 1.08, 95% CI: 0.81, 1.35, p<0.001), Hispanics with

upstate New York. Both OASIS and Medicare HH claims are publicly available datasets that can be purchased from the Centers of Medicare and Medicare Services (https://www.resdac.org/cms58fee-information-research-identifiable-data; https://www.resdac.org/file-availability); the authors had no special access privileges to the data others would not have.

**Funding:** This study was conducted with the support of the following funders: Elaine C. Hubbard Center for Nursing Research on Aging Research Endowed Award (JW), Terry Family Research Endowed Award (JW), and the Valerie and Frank Furth Fund Award (JW). The funders had no role in study design, data collection and analysis, decision to publish, or preparation of the manuscript.

**Competing interests:** The authors have declared that no competing interests exist.

dementia (β = 0.92, 95% CI: 0.38, 1.47, p = 0.001) and Asian Americans with dementia (β = 1.47, 95% CI: 0.81, 2.13, p<0.001) showed the least ADL improvement at HH discharge.

## Conclusion

Racial/ethnic minority status and dementia were associated with less ADL improvement in HH with independent and interactive effects. Policies should ensure that these patients have equitable access to appropriate, adequate community-based services to meet their needs in ADLs and disease management for improved outcomes.

## Background

Older adults with dementia often have considerable functional limitations and disease management needs with comorbidities [1, 2]. In the United States, more than 5.7 million adults older than 65 year have dementia [3], and the majority of them live at home [4]. Dementia causes gradual functional decline with increasing need for assistance with activities of daily living (ADL) over time [3]. A national study estimated that over a 5-year period, 3.3 million older Americans developed incident moderately severe dementia and had an average of three to five ADL limitations [5].

In addition, older adults with dementia experience higher prevalence of comorbidities than peers without dementia, which require substantial disease management efforts. Data have shown that 96% of older adults with dementia have at least one comorbid condition [6] and 61% have three or more comorbid chronic conditions [7], particularly heart failure, diabetes, and chronic obstructive pulmonary disease (COPD) [6, 8–10]. Comorbidities further contribute to the progression of dementia [11] and increased functional limitations which in turn decrease the self-care capacity of older adults with dementia [12]. Dementia/comorbidities and functional limitation form a downward spiral which increases the risks for hospitalizations and nursing home admissions [13–16].

Compared with older adults with dementia who live in residential or long-term nursing facilities, those who live at home are more likely to be racial/ethnic minorities, not born in the U.S., less educated, and to experience extreme poverty (annual income <$11,000) with poorer health status and more comorbidities [5]. Therefore, home and community-based services are critical to help them stay at home safely and independently.

Indeed, Medicare home-based services such as home health (HH) is received by one quarter of older adults with dementia [3]. HH provides both skilled services and non-skilled services. Skilled services include skilled nursing (SN) for disease management, physical and occupational therapy (PT, OT) that aim to improve ADL ability, and social work (SW) service to connect the patient to available resources in the community. Non-skilled services primarily refer to the homemaking assistance provided by home health aids (HA) [17]. HH has been shown to delay nursing home admission in older adults with dementia [18–24], because it reduces cognitive impairment and functional limitations, the two strongest predictors of nursing home admission [25, 26].

Yet, the effects of HH on ADL improvement has been much variant among racial/ethnic minorities and for persons living with dementia [18, 27–29]. A large body of literature documented consistent and adverse disparities among racial and ethnic minorities as compared to whites in prevalence and outcomes of dementia, including adequate use of skilled services, medications and other interventions, quality of care, and morality [30]. As people age, racial

and ethnic minorities are less likely to go to nursing homes and more likely to utilize home-based services, (e.g., HH care) than whites, highlighting the importance of home-based services to the health outcomes among minority patients [31]. Yet research suggests that racial/ethnic minorities and patients with cognitive impairment are less likely to experience ADL improvement at the end of an HH episode, as compared to non-Hispanic whites [32–35] and patients without cogntive impairment [29]. It remains unclear why such discrepant results occur. In particular, it is unknown if racial/ethnic minority status and having dementia interact with each other in their effects on functional outcome in the HH population, or if racial/ethnic minority HH recipients with or without dementia have less functional improvement than their non-Hispanic white counterparts.

The objective of this study is to examine the independent and interactive effects of minority status and dementia on ADL improvement in HH. We hypothesized that during the index HH admission, compared with non-Hispanic white patients without dementia, minorities without dementia had less ADL improvement, and minority patients with dementia had the least ADL improvement.

## Methods

### Overall design and study population

This study utilized existing data from the Outcome and Assessment Information Set (OASIS) and Medicare claims (HH claims) billing records of a non-profit HH agency in upstate New York. Both OASIS and Medicare HH claims are publicly available datasets that can be purchased from the Centers of Medicare and Medicare Services (https://www.resdac.org/cms-fee-information-research-identifiable-data; https://www.resdac.org/sites/resdac.umn.edu/files/CMS%20Fee%20List%20for%20Research%20Files_15.pdf); the authors had no special access privileges to the data others would not have.

OASIS is a standardized tool used by all Medicare-certified HH agencies to assess patient characteristics, including socioeconomic status, living arrangement, social support, health status including comorbidities, and cognitive as well as physical function. OASIS items are assessed by HH clinicians and have promising psychometric properties (coefficient alpha 0.86–0.91; Cohen's kappa 0.4–1.0; 56.3%–68.9% variance explained in factor analysis) [36–38]. In particular, OASIS assessments of ADLs and cognitive function are highly correlated with respective gold-standard measurements (cognitive function: *r = 0.62*, ADLs: *r* = 0.44 to 0.69) [37].

Inclusion criteria of the study sample included: 1) age 65 years or older; and 2) referred to HH from all sources (i.e., hospital, post-acute care and the community) and received at least one HH visit from this agency in calendar year 2017. Exclusion criteria included: 1) having missing values in key demographic variables (n = 6); 2) were not community-dwelling (i.e., nursing home residents; n = 15); 3) receiving hospice care (n = 8); 4) not having OASIS discharge assessment of ADLs due to missing data (n = 450); and 5) having an inpatient admission (n = 920), since any inpatient admission during HH can discontinue HH without completion of OASIS discharge assessment [18, 29, 39–41]. A participant flow chart was presented in Fig 1. This study was approved by the University of Rochester Medical Center Institutional Review Board with a waiver of documentation of consent due to the nature of the study (i.e., analysis of existing data).

### Variables and measures

**Dementia.** The diagnosis of dementia was determined by ICD-10 codes (documented by physicians; F01 [vascular dementia], F02-F03 [dementia in other diseases classified elsewhere

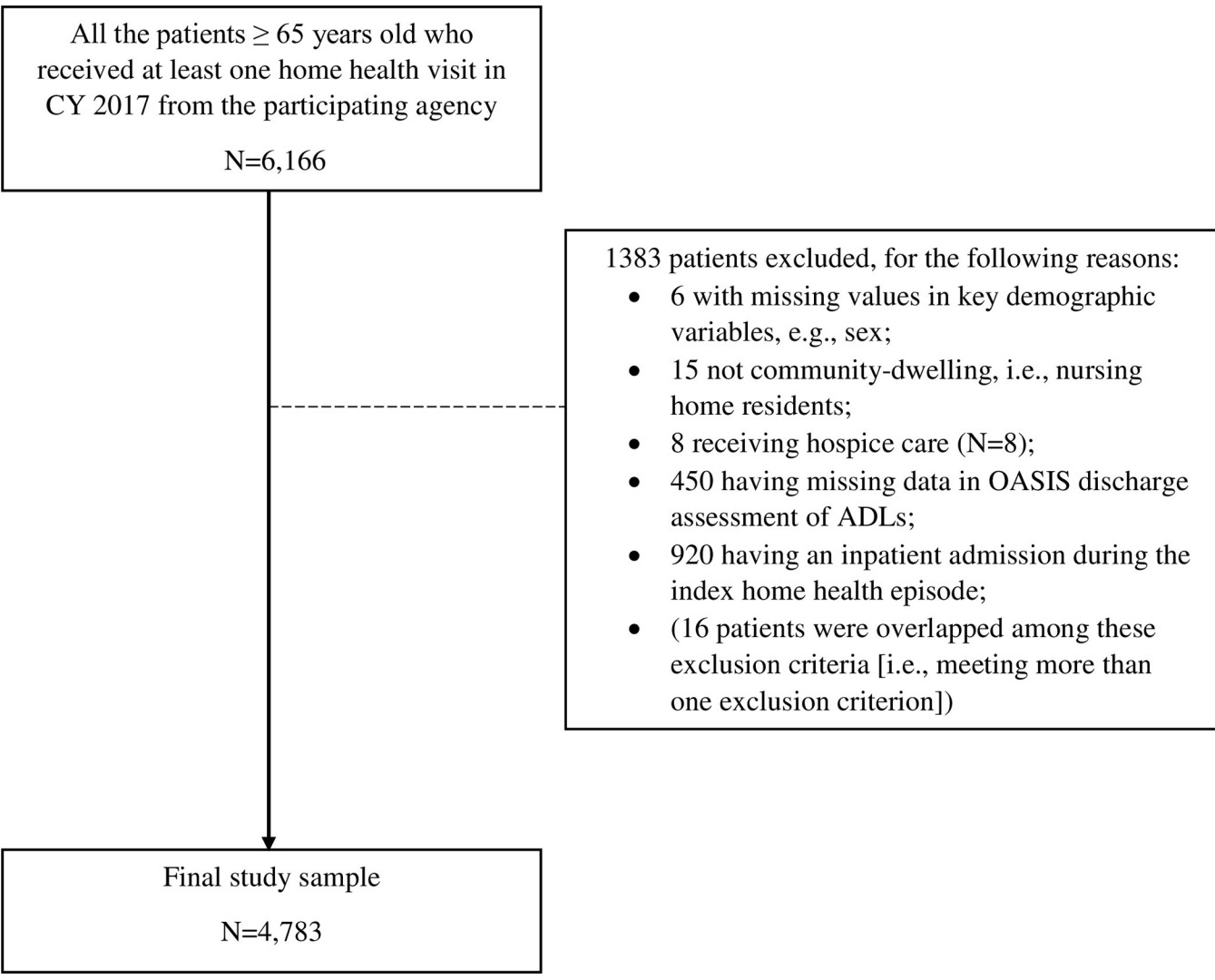

All the patients ≥ 65 years old who received at least one home health visit in CY 2017 from the participating agency

N=6,166

1383 patients excluded, for the following reasons:
- 6 with missing values in key demographic variables, e.g., sex;
- 15 not community-dwelling, i.e., nursing home residents;
- 8 receiving hospice care (N=8);
- 450 having missing data in OASIS discharge assessment of ADLs;
- 920 having an inpatient admission during the index home health episode;
- (16 patients were overlapped among these exclusion criteria [i.e., meeting more than one exclusion criterion])

Final study sample

N=4,783

**Fig 1. Participant flow chart.**

and unspecified dementia], G30 [Alzheimer's disease], G31.0 [frontotemporal dementia], G31.83 [lewy body dementia]). Because dementia is under-diagnosed in Medicare patients [42], we also used four OASIS items on cognitive functioning to supplement the identification of dementia. These items included 1) current cognitive functioning (M1700: if response was *"constant disorientation that leads to dependence more than half of the time or all the time"*), 2) frequency of confusion (M1710: if response was *"constantly or daily"*), 3) memory deficit (M1740-1: if response was *"significant memory loss that requires supervision"*), and 4) impaired decision making (M1740-2: if response was *"failure to perform usual (instrumental) activities of daily living (ADL/IADL) due to significant impairment on one's decision making ability"*). These items are assessed by direct observation of the HH clinician or via reports of the patient/caregiver on the day of assessment or within the last 14 days. Dementia was defined as having a diagnosis of dementia or cognitive impairment in OASIS (having the selected responses to any of the four OASIS items).

**Race/ethnicity.** Race and ethnicity were assessed in OASIS (M0140) in six groups, including white, African American, Hispanic, and Asian, American Indian or Alaska Native, and Native Hawaiian or Pacific Islander. Due to the small percentages of the last three groups (Asian, American Indian or Alaska Native, and Native Hawaiian or Pacific Islander), they were collapsed into one group (i.e., Other).

**ADL improvement.** he OASIS included nine ADL items (assessed by HH clinicians such as nurses and physical therapists; M1800-M1870), each on a Likert scale of 0 to N (N = maximal value; larger value indicates more limitation). The nine ADL items include grooming (0–3), dressing upper or lower body (each 0–3), bathing (0–6), toilet transferring (0–4), toilet hygiene (0–3), transferring (0–5), ambulation (0–6), and eating (0–5). Raw score of each item was standardized using the corrected Likert approach and summed to generate the composite ADL score (range: 0–9) [18, 43]. We calculated the change in the composite ADL score from HH admission to HH discharge (discharge score minus admission score), where a negative score means ADL improvement, a positive score means ADL decline, and 0 means no change.

**Covariates.** Covariates included variables related to demographic status, health status, referral source, functional limitation, and caregiver support.

Demographic status variables included age (years), sex (female/male), marital status (married/not married), living arrangement (lives alone at home/lives with others at home/ congregate settings [e.g., assisted living facilities/retirement homes]), and insurance status (Medicare-Medicaid dual eligibility [Yes/No]).

Health status was assessed by the following variables: 1) total number of diagnoses based on ICD-10 codes (range: 0–15); 2) specific diagnosis, including heart failure (I110 I11 I130 I132 I50* I09.81 I97.130 I97.131), diabetes (E08 E09 E10 E11 E13; with or without complications [E08.9 E09.9 E10.9 E11.9 E13]) and COPD (J44.9 J44.0 J44.1); 3) total number of medications, 4) obesity (M1036; Yes/No), 5) smoking (M1036; Yes/No), 6) daily interfering pain (M1242; Yes/No), and 7) current depressive symptoms (having a diagnosis of depression [ICD-10 code]) with a Patient Health Questinnaire-2 score [M1730] of $\geq$ 2 or a physician-prescribed depression intervention in the care plan [M2250]) [44, 45].

Caregiver support was defined by receiving daily assistance with ADL/IADL from informal caregivers (M2102; Yes/No).

HH service use: Because functional changes are related to the use of HH services [46, 47] and different type and volume of HH services are utilized across racial/ethnic and dementia patient groups [48–52], we adjusted for the type and volume of HH service use in the index HH admission in this study. HH service use was measured by weekly intensity of all HH services and specific HH services, including SN, PT, OT, SW, and HA. Weekly intensity was calculated as 1) number of visits/week = *([total number of visits] / [number of days receiving HHC])*7*, and 2] number of hours /week = *([total number of hours] / [number of days receiving HHC])*7*. This approach was shown as a robust measure of HH dosing in prior research [18]. Weekly intensity of total HH and specific HH services were categorized into quartiles.

## Statistical analysis

2Descriptive statistics were used to summarize sample characteristics as means (standard deviations [S.D.] or median [interquartile range]) for continuous variables and frequency (N [%]) for categorical variables. We examined the assumptions of linear regression and found that the assumptions were met (i.e., an approximately bell-curve distribution of ADL change score and studentized residuals plot; skewness = 0.296, kurtosis = 2.383). The linear regression model was performed to examine independent effect of dementia and race/

ethnicity on functional improvement, where all the covariates were entered. In the final model that tested the interactive effect of race/ethnicity and dementia on functional outcome, we defined eight groups of HH patients using the two variables, including non-Hispanic white patients without dementia (reference group), non-Hispanic white patients with dementia, African Americans without dementia, African Americans with dementia, Hispanics without dementia, Hispanics with dementia, Others (e.g. Asians) without dementia, and Others (e.g. Asians) with dementia. Statistical analyses were conducted using Stata 15.1 (College Station, TX).

## Results

The sample include 4,783 HH patients who had an average age of 80.1 years (S.D. = 9.46). The majority of the sample were non-Hispanic white (86.9%), 10.2% were African American, 1.6% were Hispanic, and 1.2% were of other minority groups (e.g. Asians). Approximately one quarter (24.2%) of the sample had dementia who had either a diagnosis of dementia (10.7%) and/or symptoms of moderate to severe cognitive impairment (22.1%). The sample had prevalent comorbidities, particularly diabetes (29.5%), heart failure (16.8%), and COPD (16.6%). At HH admission, the sample had a mean ADL limitation score of 4.1, indicating total dependency in more than four ADLs. Compared with white patients (mean = 4.1), minorities had more ADL limitations at HH admission, particularly Hispanics (mean = 4.4) and other minorities (mean 4.52) (ANOVA test: p = 0.011). The majority (83.2%) of the sample received daily caregiver assistance with ADLs. Detailed sample characteristics are presented in Table 1.

From the start to discharge of the index HH admission, the sample had a mean ADL limitation score change of -2.64 (S.D. = 1.50) with the majority (93.9%) had an ADL improvement (i.e., ADL limitation score change <0). The mean of ADL score change of the racial/ethnic and dementia cohorts was illustrated in Fig 2. Within each racial/ethnic group, patients with dementia had less ADL improvement than those without dementia at discharge from HH. The difference in ADL improvement associated with dementia was the greatest for Hispanics and other minorities (e.g., Asians). Hispanics and whites without dementia had the greatest ADL improvement, whereas African Americans and other minorities with dementia had the least ADL improvement.

As shown in Table 1, bivariate analyses showed that having an ADL improvement in HH (i.e., a negative ADL limitation score change) was significantly related to younger age, being married, having more ADL limitations at HH admission, being referred to HH after hospitalization (versus community), having a longer HH stay and receiving more hours of PT, and surprisingly, reporting daily interfering pain. On the other hand, having maintained or even declined ADL function was significantly related to racial/ethnic minority status, dementia, depressive symptoms and obesity.

The final multivariable linear regression explained 23.5% of the variance in continuous ADL improvement in HH and this model included all the covariates such as demographic status, health status, caregiver support, and HH service use (Table 2). In this model, being African American (β = 0.21, 95% confidence interval [CI]: 0.06, 0.35, p = 0.005) and having dementia (β = 0.51, 95% CI: 0.41, 0.62, p<0.001) were both independently related to less ADL improvement (i.e., positive ADL limitation score change). The interaction between racial/minority status and having dementia was associated with further decrease in ADL improvement. Specifically, African Americans with dementia (β = 1.08, 95% CI: 0.81, 1.35, p<0.001), Hispanics with dementia (β = 0.92, 95% CI: 0.38, 1.47, p = 0.001), and other minorities (e.g., Asians) with dementia (β = 1.47, 95% CI: 0.81, 2.13, p<0.001) showed the least ADL improvement at

**Table 1. Sample characteristics.**

| Variable | Entire Sample (4,783) | Functional Improvement Cohorts | | |
| --- | --- | --- | --- | --- |
| | | Same/Worsened (290, 6.1%) | Improved (4,493, 93.9%) | p-value* |
| Age, mean (S.D.) | 80.1 (9.46) | 81.4 (10.36) | 80.0 (9.40) | 0.007 |
| Female, N(%) | 2820 (58.9%) | 178 (61.4%) | 2642 (58.8%) | 0.39 |
| Race/ethnicity, N(%) | | | | |
| • Non-Hispanic white | 4159 (86.9%) | 235 (81.0%) | 3924.3%) | |
| • African American | 489 (10.2%) | 44 (15.2%) | 445..9%) | |
| • Hispanic | 77 (1.6%) | 6 (2.1%) | 71 1.6%) | |
| • Other | 58 (1.2%) | 5 (1.7%) | 53 (1.2%) | 0.021 |
| Dementia, N(%) | 1156 (24.2%) | 135 (46.6%) | 1021 (22.7%) | <0.001 |
| Married, N(%) | 2147 (44.9%) | 114 (39.3%) | 2033 (45.3%) | 0.049 |
| Living arrangement, N(%) | | | | |
| • Living with others | 2892 (60.5%) | 178 (61.4%) | 2714.4%) | |
| • Living alone | 1385 (28.9%) | 77 (26.6%) | 1308.1%) | |
| • Congregated settings | 506 (10.6%) | 35 (12.1%) | 471 (10.5%) | 0.52 |
| Number of diagnoses, mean (S.D.) | 8.9 (3.42) | 8.9 (3.47) | 8.9 (3.42) | 0.57 |
| Number of medications, mean (S.D.) | 12.8 (5.42) | 11.8 (5.47) | 12.8 (5.42) | 0.99 |
| Depressive symptoms, N(%) | 2170 (45.4%) | 161 (55.5%) | 2009 (44.7%) | <0.001 |
| Obesity, N(%) | 1060 (22.2%) | 48 (16.6%) | 1012 (22.5%) | 0.018 |
| Current Smoking, N(%) | 860 (17.9%) | 64 (22.1%) | 796 (17.7%) | 0.06 |
| Daily interfering pain, N(%) | 3410 (71.3%) | 166 (57.2%) | 3244 (72.2%) | <0.001 |
| Heart Failure, N(%) | 802 (16.8%) | 43 (14.8%) | 759 (16.9%) | 0.36 |
| Diabetes without complications, N(%) | 850 (17.8%) | 48 (16.6%) | 802 (17.9%) | 0.58 |
| Diabetes with complications, N(%) | 561 (11.7%) | 42 (14.5%) | 519 (11.6%) | 0.13 |
| Chronic obstructive pulmonary disease, N(%) | 794 (16.6%) | 38 (13.1%) | 756 (16.8%) | 0.09 |
| Baseline ADL limitation score, mean (S.D.) | 4.1 (1.19) | 4.2 (2.09) | 4.1 (1.11) | 0.03 |
| Receiving daily caregiving assistance with ADL, N(%) | 3980 (83.2%) | 232 (80.0%) | 3748 (83.4%) | 0.13 |
| Referral source | | | | <0.001 |
| • After hospitalization | 2974 (62.2%) | 149 (51.4%) | 2825.9%) | |
| • After post-acute care | 601 (12.6%) | 20 (6.9%) | 581 2.9%) | |
| • Community | 1208 (25.3%) | 121 (41.7%) | 1087 (24.2%) | |
| Days of index home health admission (median, $Q_1$, $Q_3$) | 31 (20, 49) | 24.5 (14, 44) | 32 (21, 49) | <0.001 |
| Index home health admission > 60 days | 439 (9.2%) | 21 (7.2%) | 418 (9.3%) | 0.24 |
| Index home health admission > 120 days | 100 (2.1%) | 6 (2.1%) | 94 (2.1%) | 0.98 |
| Total hours of skilled nursing, mean (S.D.) | 4.6 (5.16) | 5.1 (4.62) | 4.5 (5.19) | 0.095 |
| Total hours of physical therapy, mean (S.D.) | 4.7 (5.16) | 2.5 (4.12) | 4.9 (5.19) | <0.001 |
| Total hours of occupational therapy, mean (S.D.) | 2.8 (105.3) | 0.8 (2.58) | 2.9 (108.65) | 0.735 |
| Total hours of aide assistance (homemaking), mean (S.D.) | 1.4 (13.06) | 2.0 (8.78) | 1.3 (13.28) | 0.394 |
| Total hours of social work, mean (S.D.) | 0.1 (0.38) | 0.1 (0.43) | 0.1 (0.38) | 0.043 |

*p-value based on t-test for continuous variables and Chi-Square test for categorical variables.

HH discharge among all racial/ethnic and dementia groups. This multivariable model also showed that receiving more hours of PT (β = - 0.03, 95% CI: - 0.04, - 0.02, p<0.001), having more ADL limitation at HH admission and living alone were also related to greater ADL improvement in HH, whereas heart failure and depressive symptoms, and referral source from non-hospital settings were related to less ADL improvement.

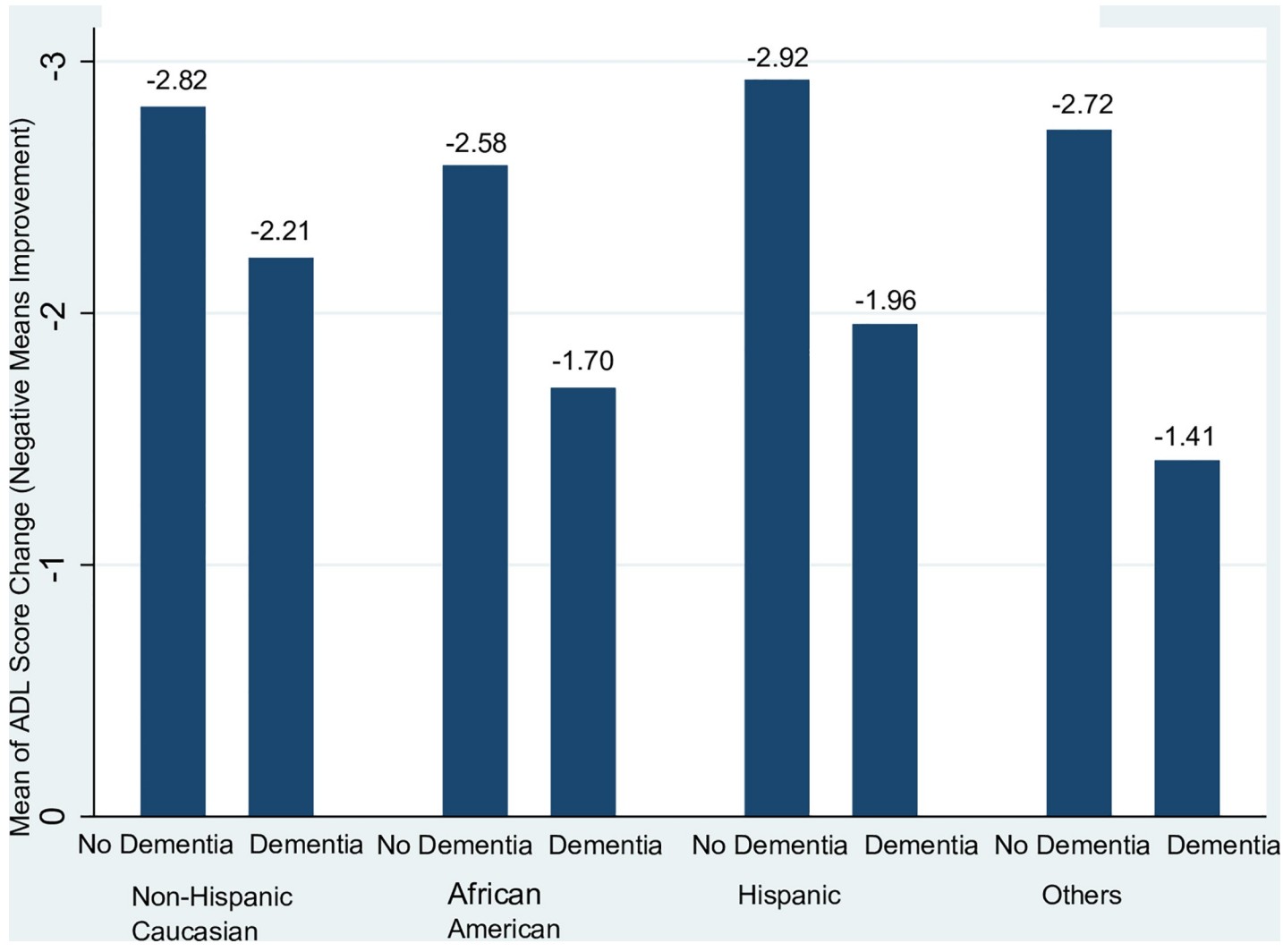

**Fig 2. Mean of ADL score change in racial/ethnic and dementia cohorts.**

## Discussion

To our knowledge, this is the first study that examined the independent and interactive effects of dementia and minority status on ADL improvement in HH, with three principal findings. First, both racial/ethnic minority status and having dementia were independently related to less ADL improvement in HH. Second, significant interaction effects on less ADL improvement were found between dementia and being African Americans, Hispanics, or other racial/ethnic minorities such as Asian Americans. Compared with non-Hispanic whites without dementia, minority patients with dementia had the least ADL improvement after adjusting for patient covariates and variation in the type and volume of received HH services. Finally, receiving more PT service was related to increased ADL improvement.

Our study confirmed previous findings on the independent associations of cognitive impairment [29, 53] and racial/ethnic minority status [32, 34] with reduced degrees of ADL improvement in HH. In addition, this study revealed significant interaction between racial/ethnic minority and dementia that was related to further decrease in ADL improvement in HH. This finding has four plausible explanations.

**Table 2. Multivariable linear regression of ADL improvement in HH.**

| ADL score change (score negative: improvement; positive: increase) | Regression coefficient β | P-value | 95% Confidence Interval | |
|---|---|---|---|---|
| Race/ethnicity and dementia interaction (reference: white without dementia) | | | | |
| • African American, no dementia | 0.207 | 0.005 | 0.062 | 0.351 |
| • Hispanic, no dementia | 0.026 | 0.887 | -0.332 | 0.384 |
| • Asian, no dementia | 0.018 | 0.929 | -0.387 | 0.424 |
| • white, dementia | 0.513 | <0.001 | 0.408 | 0.619 |
| • African American, dementia | 1.077 | <0.001 | 0.805 | 1.349 |
| • Hispanic, dementia | 0.922 | 0.001 | 0.375 | 1.469 |
| • Asian, dementia | 1.469 | <0.001 | 0.811 | 2.128 |
| Medicare-Medicaid Dual eligibility (reference: No) | 0.175 | 0.198 | -0.091 | 0.442 |
| Age | 0.019 | <0.001 | 0.014 | 0.023 |
| Female (reference: male) | -0.014 | 0.739 | -0.095 | 0.068 |
| Married (reference: not married) | -0.090 | 0.058 | -0.182 | 0.003 |
| Living arrangement (reference: living with others at home) | | | | |
| • Live alone | -0.335 | <0.001 | -0.441 | -0.229 |
| • Assisted living facilities | 0.113 | 0.118 | -0.029 | 0.255 |
| Referral source (reference: after hospitalization) | | | | |
| • After post-acute care | 0.147 | 0.016 | 0.027 | 0.267 |
| • Community | 0.683 | <0.001 | 0.588 | 0.779 |
| Number of diagnoses | 0.014 | 0.042 | 0.001 | 0.027 |
| Number of medications | -0.001 | 0.732 | -0.009 | 0.006 |
| Heart failure (Yes/No) | 0.123 | 0.025 | 0.015 | 0.230 |
| COPD (Yes/No) | -0.119 | 0.034 | -0.229 | -0.009 |
| Diabetes without complications (Yes/No) | -0.008 | 0.875 | -0.112 | 0.095 |
| Diabetes with complications (Yes/No) | 0.071 | 0.280 | -0.058 | 0.201 |
| Obesity (Yes/No) | -0.014 | 0.776 | -0.109 | 0.081 |
| Smoking (current or past; Yes/No) | -0.058 | 0.284 | -0.164 | 0.048 |
| Daily interfering pain (Yes/No) | -0.300 | <0.001 | -0.388 | -0.213 |
| Depressive symptoms (Yes/No) | 0.162 | <0.001 | 0.081 | 0.243 |
| Receiving daily caregiver assistance | -0.046 | 0.433 | -0.161 | 0.069 |
| Baseline functional limitation | -0.395 | <0.001 | -0.430 | -0.360 |
| Service use during the index home health admission | | | | |
| • Total hours of skilled nursing | 0.006 | 0.105 | -0.001 | 0.014 |
| • Total hours of physical therapy | -0.030 | <0.001 | -0.037 | -0.022 |
| • Total hours of occupational therapy | 0.000 | 0.322 | -0.001 | 0.000 |
| • Total hours of aide assistance (homemaking) | 0.002 | 0.311 | -0.001 | 0.004 |
| • Total hours of social work | 0.169 | 0.001 | 0.068 | 0.271 |

First, dementia may have reduced the responsiveness of functional limitation to intervening HH services such as PT. Different from functional decline introduced by acute illness or injury that is partially reversible [54], functional decline due to Alzheimer's disease is progressive and less responsive to interventions [55]. Racial/ethnic minority patients, particularly African Americans and Hispanics, are more likely than whites to have a diagnosis of dementia [30, 56]. Therefore, it is likely that a larger proportion of the ADL limitation among racial/ethnic minorities is related to dementia and less likely to improve during HH.

Second, the differences in unmet needs related to chronic disease management among different racial/ethnic groups may as well contribute to the interactive effect of racial/ethnic minority status and dementia leading to less ADL improvement. It is worth mentioning that

racial/ethnic minority patients had more baseline ADL limitations than white patients at HH admission (mean ADL limitations: 4.09 [white], 4.36 [Hispanics], and 4.52 [Others]; p-value of ANOVA test = 0.011), which suggests that 1) minority patients may have more opportunity to have improved ADL function at HH discharge if they receive adequate support and services at home, and that 2) HH may need to be provided earlier when the patient's physical function is less impaired. These alternative hypotheses need to be tested in future studies.

However, racial/ethnic minorities may have limited access to healthcare resources, which makes it difficult for them to improve in ADLs. Substantial evidence exists on the consistent and adverse racial/ethnic disparities in the incidence, prevalence, outcomes and mortality of dementia [30, 56, 57], which is likely due to the inadequate receipt of intervention, unsatisfactory management of chronic comorbidities and insufficient use of health care service among racial/ethnic minority patients [56], such as HH service. Research has shown that among patients with dementia, racial/ethnic minorities have lower socioeconomic status and therefore fewer resources to meet the needs of general health care and daily activities [58]; and these unmet needs in health care and particularly chronic disease management may have hindered the progress of functional improvement among racial/ethnic minorities. Similarly, in post hoc analysis of this study, we found that the rate of Medicare and Medicaid dual eligibility was higher among Asians (15.5%), African Americans (7.2%) and Hispanics (6.5%) than whites (1.3%), suggesting limited financial resources of racial/ethnic minority families. We used Medicare and Medicaid dual eligibility status (reported in the OASIS) as a proxy of socioeconomic status, knowing that it only represents one aspect of socioeconomic status. Future research is needed to more comprehensively measure socioeconomic status with detailed education and income data when available.

The third explanation is related to the differences in caregiver support. The Alzheimer's Disease Association has estimated that in 2017, over 16 million family and other unpaid caregivers of people with Alzheimer's disease or other dementia provided 18.4 billion hours of unpaid care, averaging to approximately 88 hours of care per caregiver per 30 days [3]. We conducted ad hoc analyses and found that patients with dementia in this study received an average of 12.9 hours of HH services over the entire HH admission (i.e., 31 days). As such, among patients with dementia and chronic conditions, having high-quality caregiver support is key to successfully retaining the effects of intermittent, short-term HH interventions [19, 59]. Although we adjusted for the receipt of daily caregiver support, we did not have detailed information on the type, content, and amount of caregiver support received by the patients in this study. Given that racial/ethnic minority families have different structures and sources of social support [60], and minority caregivers often have different preferences and beliefs about their roles in providing care for patients with dementia [61], we could not rule out the possibility that unmeasured variance in family and social support structure, particularly the degrees of caregiver support and family cohesion may have contributed to the racial/ethnic difference in ADL improvement during HH.

Last but not least, this finding of lower ADL improvement among patients with dementia may be attributed to the challenge in providing HH care to patients with dementia that is intensified by the cultural and/or language barriers related to the patient's racial/ethnic minority status. For example, effective communication and identification of unmet needs with symptom management are essential care processes in HH [59]. These care processes may be more difficult to realize in the presence of dementia or cognitive impairment [62], especially given the inadequate training and confidence of many HH clinicians to provide targeted care for persons with dementia [63]. Indeed, studies on Medicare HH patients showed that those with a dementia diagnosis consistently reported lower satisfaction with the HH care that they received, along several assessed dimensions such as "communication between providers and

patient", and "specific care issues" such as management of medication, pain, and safety [64]. In addition, racial/ethnic minorities often find it hard to describe symptoms to language-discordant clinicians that further affect the quality and outcomes of care they receive [65–68]. Therefore, it is likely that the communication difficulty with patients with dementia compounded by the language barrier of racial/ethnic minorities have affected the quality of HH and its effectiveness at improving ADL function.

We found that patients receiving more PT showed more ADL improvement, which is consistent with existing literature. For example, a study analyzing national OASIS data of Medicare beneficiaries with heart failure showed that patients who used any PT were more likely to have improved ADL function [18]. Although dementia pathology can cause substantial functional decline, research has shown that rehabilitation interventions are equally effective at improving physical function between older adults with dementia and those without dementia [69]. However, it is of question if patients with dementia should receive intensive PT services, as too much PT may cause more harm than benefit for critically ill patients [70]. To date, there is no consensus on the adequate and optimal amount of therapy services that are needed for pronounced ADL improvement, given the heterogeneity in demographic, clinical, functional, and social support profiles among HH recipients. On January 01, 2020, the Centers for Medicare and Medicaid Services will implement a new case-mix classification model, i.e., the Patient-Driven Groupings Model (PDGM) [71]. The PDGM places less emphasis on therapy services and will likely decrease the amount of PT received in the entire Medicare HH population. As such, it is critical that an appropriate mix and amount of HH services are offered to HH patients who are at risk for less ADL improvement during HH, i.e., minorities with dementia.

In addition to PT, HH provide services to help offset the negative impact of functional decline on chronic disease management (SN), household tasks (HA), and social engagement (SW). In this study, we did not find the volume of SN and HA use to be significantly associated with ADL improvement, as shown in previous studies. A study analyzing OASIS records of an urban HH agency showed that patients receiving a higher continuity of HA were more likely to improve in ADLs [41]. Another study with 107 HH patients with heart failure showed that those receiving more SN visits per week had a higher degree of ADL improvement [39]. Such discrepancy in findings can be explained by the differences between our study and earlier studies in the samples (i.e., patients with heart failure; urban population with high concentration of low-income patients [vs. patients with all diagnoses in both urban and rural regions in this study]) and unit of analysis (i.e., average visits per week; continuity [vs. total number of hours in this study], and therefore does not negate the effect of non-PT services in HH on improving the self-care ability of patients while living at home.

In 2014, the Centers for Medicare and Medicaid Services passed the Improving Medicare Post-Acute Care Transformation (IMPACT) act that standardized physical function assessment across all post-acute care settings to facilitate care coordinate and improve patient outcomes [72]. One third of facility-based post-acute care referrals are refused due to the patient's preference of staying at home [73], emphasizing the need of adequate home-based support for older adults following an acute hospital stay. However, post-acute patients are not the only population for whom maintaining adequate physical function is critical. Our results showed that patients who are referred from the community, as compared to recently hospitalized and post-acute patients, had less ADL improvement at HH discharge. In particular, racial/ethnic minorities with dementia are among the groups with the least ADL improvement at the discharge from HH. Because the majority of patients with dementia live in the community [3], and over 60% have three or more comorbidities [2], receiving intervening services for better

physical function is fundamental to effective management of comorbidities and the patient's outcomes as well as ability of living at home as they desire.

Strengths of this study include: 1) specific examination of the interaction between racial/ ethnic minority status and dementia in functional outcomes in HH; 2) using administrative billing data to assess HH service use; and 3) using multiple data sources to identify dementia by the diagnoses (administrative data) and cognitive impairment symptoms (OASIS).

The study has a few limitations. First, we used data from one HH agency, so findings may not be generalizable to HH agencies in other regions. Second, despite the satisfactory validity and reliability of OASIS items in assessing cognitive and physical function [36–38, 74, 75] and their widespread use in research [29, 76, 77], the four OASIS items on cognitive functioning we used to identify the cognitive impairment could not differentiate dementia from other common geriatric psychiatric conditions such as delirium and depression. Third, the findings are limited by the lack of information about the severity of dementia in the OASIS or Medicare claims (where the billing records were extracted), thus we could not differentiate whether dementia stage and severity of behavior and cognitive impairments play an important role in functional outcomes. Because racial/ethnic minority persons tend to have fewer contacts with health care providers than white older adults, minority HH patients may have a diagnosis of dementia when it is at a more advanced stage. Thus, the potentially more severe dementia among racial/ethnic minorities may underlie partially the findings of this study. Moreover, since we used the OASIS cognitive status to help identify patients with dementia, we do not have a diagnosis of the specific type of dementia for these cases (unless they also had an ICD-10 diagnosis for dementia). Future research could combine multiple clinical and administrative datasets that contain accurate information about clinical diagnosis, dual eligibility status, functional status, and severity of dementia for more refined analyses. Fourth, OASIS does not have information on education, which is closely related to one's cognitive function and risk of dementia [78], but is also an important measure of the socioeconomic status of the patient. Last, we did not have data on the gradient of caregiver support, such as the role of caregivers, the type, amount, content and quality of caregiver support. Future research needs to examine how HH providers should engage caregivers for better outcomes of HH recipients.

## Conclusion

Among Medicare HH recipients, racial/ethnic minority patients with dementia improved the least in ADL function at HH discharge. Future studies should examine the unmet needs among racial/ethnic minority patients who are living at home with dementia. In face of the upcoming PDGM that places less emphasis on therapy services, policies should ensure that these patients have equal access to appropriate, adequate community-based services to maintain and improve ADL ability.

## Author Contributions

**Conceptualization:** Jinjiao Wang, Yue Li.

**Data curation:** Jinjiao Wang, Thomas V. Caprio.

**Formal analysis:** Jinjiao Wang, Xueya Cai, Yue Li.

**Investigation:** Jinjiao Wang.

**Methodology:** Jinjiao Wang, Fang Yu, Xueya Cai, Yue Li.

**Project administration:** Jinjiao Wang.

**Resources:** Jinjiao Wang, Thomas V. Caprio.

**Software:** Jinjiao Wang.

**Supervision:** Jinjiao Wang, Yue Li.

**Validation:** Jinjiao Wang.

**Visualization:** Jinjiao Wang.

**Writing – original draft:** Jinjiao Wang.

**Writing – review & editing:** Jinjiao Wang, Fang Yu, Xueya Cai, Thomas V. Caprio, Yue Li.

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
