## [Decision Letter · Decision Letter 0]

9 Jan 2020

PONE-D-19-29178

Functional Outcome in Home Health: Do minority patients with dementia fare worse?

PLOS ONE

Dear Dr. Wang,

Thank you for submitting your manuscript to PLOS ONE. After careful consideration, we feel that it has merit but does not fully meet PLOS ONE’s publication criteria as it currently stands. Therefore, we invite you to submit a revised version of the manuscript that addresses the points raised during the review process.

ACADEMIC EDITOR: Please develop a flow diagram of sample selection criteria, including inclusion and exclusion criteria. The results begin in the present tense, but then switch to the past tense. Make sure the results is written in the past tense throughout.

We would appreciate receiving your revised manuscript by Feb 23 2020 11:59PM. To enhance the reproducibility of your results, we recommend that if applicable you deposit your laboratory protocols in protocols.io, where a protocol can be assigned its own identifier (DOI) such that it can be cited independently in the future. For instructions see: http://journals.plos.org/plosone/s/submission-guidelines#loc-laboratory-protocols

We look forward to receiving your revised manuscript.

Kind regards,

Nikki R. Wooten, PhD

Academic Editor

PLOS ONE

Journal Requirements:

2. Please note that outmoded terms and potentially stigmatizing labels should be changed to more current, acceptable terminology. Examples: “Caucasian” should be changed to “white” or “of [Western] European descent” (as appropriate); “cancer victims” should be changed to “patients with cancer.” https://journals.plos.org/plosone/s/submission-guidelines#loc-human-subjects-research

3. Please modify the title to ensure that it is meeting PLOS’ guidelines (https://journals.plos.org/plosone/s/submission-guidelines#loc-title). In particular, the title should be "specific, descriptive, concise, and comprehensible to readers outside the field". We recommend modifying the title as "Functional Outcome in Home Health: Do ethnic minority patients with dementia fare worse?

4. In ethics statement in the manuscript and in the online submission form, please provide additional information about the patient records used in your retrospective study. Specifically, please ensure that you have discussed whether all data were fully anonymized before you accessed them and/or whether the IRB or ethics committee waived the requirement for informed consent. If patients provided informed written consent to have data from their medical records used in research, please include this information.

Please provide an amended Funding Statement that declares *all* the funding or sources of support received during this specific study (whether external or internal to your organization) as detailed online in our guide for authors at http://journals.plos.org/plosone/s/submit-now.  Please state what role the funders took in the study.  If any authors received a salary from any of your funders, please state which authors and which funder. If the funders had no role, please state: "The funders had no role in study design, data collection and analysis, decision to publish, or preparation of the manuscript."

6. We note that you have indicated that data from this study are available upon request. PLOS only allows data to be available upon request if there are legal or ethical restrictions on sharing data publicly. For information on unacceptable data access restrictions, please see http://journals.plos.org/plosone/s/data-availability#loc-unacceptable-data-access-restrictions.

Reviewers' comments:

Reviewer's Responses to Questions

**Comments to the Author**

1. Is the manuscript technically sound, and do the data support the conclusions?

Reviewer #1: Yes

Reviewer #2: Yes

2. Has the statistical analysis been performed appropriately and rigorously? 

Reviewer #1: Yes

Reviewer #2: No

3. Have the authors made all data underlying the findings in their manuscript fully available?

Reviewer #1: Yes

Reviewer #2: Yes

4. Is the manuscript presented in an intelligible fashion and written in standard English?

Reviewer #1: Yes

Reviewer #2: Yes

5. Review Comments to the Author

Reviewer #1: This is the first submission of an article titled “Functional Outcomes in Home health: Do minority patients with dementia fare worse”. The authors are a team or researchers with expertise in nursing, biostatistics, geriatrics, and public health. Study purpose is to examine independent and interactive effects of dementia with racial/ethnic minority status on outcomes during home health admissions for those with Medicare. The study is a retrospective analysis of OASIS records from an agency in NY. The team proposes that the effects of home health on activity of daily living improvement is variable among racial/ethnic minorities and those with dementia. They hypothesized that during home health admission, minority patients with dementia will have less ADL improvement than non-Hispanic Caucasian patients. They used data from non-profit HH agencies to assess patient characteristics, socioeconomic status, living arrangement, social support, health status, and cognitive and physical function. The team reports that Hispanics and Caucasians without dementia had the greatest ADL improvement, while individuals who are African American or from other minorities with dementia had the least ADL improvement. I found these findings to be interesting – but do note some concerns regarding limited discussion of alternative viable hypotheses, consideration for education, consideration for severity of dementia, consideration for social economic status discrepancies.

Page 3: Inclusion criteria; I suggest participant design flow to show the method by which patient data were extracted. This will be thorough.

Page 3: incomplete sentence “This study was approved by the University of XXX Institutional review board. “ The university is not named here.

Page 3: Diagnosis of dementia is via ICD-10 codes – but who diagnosed the dementia as well as the ADL status.

Major limitation: Cognitive function is not actually assessed, from my observation. It appears to be an opinion based on the data field entered in from the physician – but unknown how this was determined. Is this based on objective testing, patient family feedback, interviews, or simple behavioral observation?

Major limitation: Rationale for the hypothesis is unclear. The opposite pattern could be hypothesized – i.e., individuals with racial/ethnic diversity may have more improvement in ADL due to more cohesive families, perhaps. I encourage the team to clarify why racial/ethnic networks would be more vulnerable than individuals who are non-Hispanic/ Caucasian.

Major limitation is that education is not considered in this model.

Major limitation is that although the database assumes access to social economic status – I do not see SES mentioned in the table or the model outcome. It is mentioned once in the paper – and that is in the reference to the OASIS record. Can the authors please clarify if these data are able to be acquired?

Major limitation: The authors appear to assume the dementias are at the same level of severity. It is unknown if the African American minority patients were at a greater level of dementia – and were taken to the physician at a later timepoint due to cultural or family factors. This would explain the limited improvement. Also, type of dementia is not considered.

Reviewer #2: This manuscript addresses an important topic with several key implications for the management of dementia in those receiving home care. It highlights racial disparity in outcomes of older adults receiving home care. The authors have used a rich data source that was well-described for the analysis. However, I have some reservations about the study’s conclusions because of the analysis. These are listed below

1. The authors used multiple linear regression for the analysis without examining whether the assumptions of linear regression are satisfied in the data. More specifically, the assumption of normal distributions was not examined. The authors should present the distribution of the change scores and the corresponding regression residuals.

2. A distribution of the change scores should be presented along with an estimate of the effect size

3. I am surprised that the authors did not examine the direct effect of ethnicity and dementia on functional outcomes but only reported the ethnicity-dementia interactive effects. The authors should revise the analysis to include ethnicity and dementia as main effects in addition to the interaction effects.

6. PLOS authors have the option to publish the peer review history of their article (what does this mean?). If published, this will include your full peer review and any attached files.

Reviewer #1: No

Reviewer #2: Yes: Tolulope Sajobi

---

## [Author Response · Author response to Decision Letter 0]

4 Mar 2020

Please see our response to the reviewer and editor comments in the attached file "Response to Reviewers comments".

---

## [Decision Letter · Decision Letter 1]

11 May 2020

Functional Outcome in Home Health: Do Racial and Ethnic Minority Patients with Dementia Fare Worse?

PONE-D-19-29178R1

Dear Dr. Wang,

We are pleased to inform you that your manuscript has been judged scientifically suitable for publication and will be formally accepted for publication once it complies with all outstanding technical requirements.

With kind regards,

Luisa N. Borrell, DDS, PhD

Academic Editor

PLOS ONE

Additional Editor Comments (optional):

You have addressed the reviewers' comments and suggestions satisfactorily.

Reviewers' comments:

Reviewer's Responses to Questions

**Comments to the Author**

1. If the authors have adequately addressed your comments raised in a previous round of review and you feel that this manuscript is now acceptable for publication, you may indicate that here to bypass the “Comments to the Author” section, enter your conflict of interest statement in the “Confidential to Editor” section, and submit your "Accept" recommendation.

Reviewer #2: All comments have been addressed

2. Is the manuscript technically sound, and do the data support the conclusions?

Reviewer #2: Yes

3. Has the statistical analysis been performed appropriately and rigorously? 

Reviewer #2: Yes

4. Have the authors made all data underlying the findings in their manuscript fully available?

Reviewer #2: No

5. Is the manuscript presented in an intelligible fashion and written in standard English?

Reviewer #2: Yes

6. Review Comments to the Author

Reviewer #2: (No Response)

7. PLOS authors have the option to publish the peer review history of their article (what does this mean?). If published, this will include your full peer review and any attached files.

Reviewer #2: Yes: Tolulope Sajobi

---

## [Editor Report · Acceptance letter]

13 May 2020

PONE-D-19-29178R1 

Functional Outcome in Home Health: Do Racial and Ethnic Minority Patients with Dementia Fare Worse? 

Dear Dr. Wang:

I am pleased to inform you that your manuscript has been deemed suitable for publication in PLOS ONE. Congratulations! Your manuscript is now with our production department. 

With kind regards,

on behalf of

Dr. Luisa N. Borrell 

Academic Editor

PLOS ONE